# Weekends-Off Lenvatinib for Unresectable Hepatocellular Carcinoma Improves Therapeutic Response and Tolerability Toward Adverse Events

**DOI:** 10.3390/cancers12041010

**Published:** 2020-04-19

**Authors:** Hideki Iwamoto, Hiroyuki Suzuki, Shigeo Shimose, Takashi Niizeki, Masahito Nakano, Tomotake Shirono, Shusuke Okamura, Yu Noda, Naoki Kamachi, Toru Nakamura, Atsutaka Masuda, Takahiko Sakaue, Toshimitsu Tanaka, Dan Nakano, Miwa Sakai, Taizo Yamaguchi, Ryoko Kuromatsu, Hironori Koga, Takuji Torimura

**Affiliations:** 1Division of Gastroenterology, Department of Medicine, Kurume University School of Medicine, Kurume 830-0011, Japan; suzuki_hiroyuki@med.kurume-u.ac.jp (H.S.); shimose_shigeo@med.kurume-u.ac.jp (S.S.); nakano_masahito@med.kurume-u.ac.jp (M.N.); shirono_tomotake@med.kurume-u.ac.jp (T.S.); okamura_shyuusuke@kurume-u.ac.jp (S.O.); noda_yuu@med.kurume-u.ac.jp (Y.N.); kamachi_naoki@med.kurume-u.ac.jp (N.K.); masuda_atsutaka@med.kurume-u.ac.jp (A.M.); sakaue_takahiko@med.kurume-u.ac.jp (T.S.); tanaka_toshimitsu@med.kurume-u.ac.jp (T.T.); nakano_dan@med.kurume-u.ac.jp (D.N.); sakai_miwa@med.kurume-u.ac.jp (M.S.); ryoko@med.kurume-u.ac.jp (R.K.); hirokoga@med.kurume-u.ac.jp (H.K.); tori@med.kurume-u.ac.jp (T.T.); 2Iwamoto Internal Medicine Clinic, Kitakyusyu 802-0832, Japan; sota-haruki@hotmail.co.jp

**Keywords:** lenvatinib, hepatocellular carcinoma, weekends-off, adverse events, molecular targeted agents

## Abstract

*Background:* Although lenvatinib has become the standard therapy for hepatocellular carcinoma (HCC), the high incidence rate of adverse events (AEs) is an issue. This study aimed to clarify the AEs of lenvatinib and the therapeutic impact of five days-on/two days-off administration (i.e., weekends-off strategy) for lenvatinib. *Methods*: We retrospectively assessed the therapeutic effects and AEs of 135 patients treated with lenvatinib, and the improvement of tolerability and therapeutic efficacy of 30 patients treated with the weekends-off strategy. We also evaluated lenvatinib-induced vascular changes in tumors and healthy organs using a mouse hepatoma model. *Results*: The incidence rates of any grade and grade ≥ 3 AEs were 82.1% and 49.6%. Fatigue was the most important AE since it resulted in dose reduction and discontinuation. Of the 30 patients who received weekends-off lenvatinib, 66.7% tolerated the AEs. Although 80.8% of the patients showed progression after dose reduction, the therapeutic response improved in 61.5% of the patients by weekends-off lenvatinib. Notably, weekends-off administration significantly prolonged the administration period and survival (*p* < 0.001 and *p* < 0.05). The mouse hepatoma model showed that weekends-off administration contributed to recovery of vascularity in the organs. *Conclusion*: Weekends-off administration of lenvatinib was useful to recover the therapeutic response and tolerability toward AEs.

## 1. Introduction

Hepatocellular carcinoma (HCC) is the fifth most common cancer and is the third leading cause of cancer-related death worldwide [1]. In 2018, the World Health Organization (WHO) reported that more than 0.7 million patients worldwide died of HCC. This is partly because patients with HCC are often diagnosed at an advanced stage [2]. Several treatment guidelines recommend molecular-targeting agents (MTAs) for advanced HCC [3,4,5]. Therefore, MTAs have been increasingly used as the standard therapy for patients with advanced HCC. Sorafenib was the first approved MTA and has been proven to have a survival benefit in two randomized phase III clinical trials [6,7]. Various MTAs have since been approved as the first-line and second-line drugs for HCC more than a decade after sorafenib was first approved [8,9]. Lenvatinib is one of the MTAs approved as a first-line drug for unresectable HCC in the United States, the European Union (EU), and Asian countries. The phase III REFLECT trial has shown noninferiority of lenvatinib to sorafenib with respect to overall survival (OS) [10], but there were significant differences in progression-free survival (PFS), time to progression, and an objective response rate (ORR). Despite the apparent therapeutic benefit of MTAs for advanced HCC, the high rate of adverse events (AEs) currently limits its application [11]. Approximately 80%–90% of patients who were treated with sorafenib or lenvatinib developed a grade of AEs, and more than 50% of patients who develop grade 3 AEs need dose reduction or treatment discontinuation. Therefore, the management of MTA-induced AEs is crucial for its long-term administration.

The current MTAs approved for the treatment of HCC are those with an anti-angiogenic property targeting vascular endothelial growth factor (VEGF) [12]. We have previously reported about the mechanisms by which MTA caused AEs. MTAs inhibited not only tumor angiogenesis, but also damaged the vascular structure of other healthy organs, such as the thyroid, adrenal gland, and gastrointestinal tract [13,14,15]. Although various supportive medicines that reduce AEs have been developed, their effects are limited [16,17,18]. Furthermore, the optimal management for MTA-induced AEs has not been established. In cases of severe AEs during MTA treatment, the dose is reduced or the treatment is interrupted. However, many patients experience tumor revascularization and regrowth after dose reduction (Appendix A). A possible method for AE management aside from the use of supportive medicines is adjustment of the administration schedule. In general, it is recommended to adminster lenvatinib daily and consecutively. Ikeda et al. [19] and Tamai et al. [20] reported that the peak concentration (Cmax) and area under the blood concentration-time curve (AUC) is important for the therapeutic effects of lenvatinib. The Cmax and AUC of lenvatinib are thought to be correlated with the anti-tumor effect and appearance of AEs in lenvatinib, respectively. However, it is difficult to maintain the Cmax and AUC through gradual dose reduction alone. According to Ikeda et al.’s report, reducing the dose of lenvatinib from 12 mg to 8 mg led to a 48% decrease in Cmax and a 45% decrease in the AUC. Therefore, we considered it important to refine the administration schedule in order to maintain the Cmax and the AUC. Hence, we developed the “weekends-off” administration method with which lenvatinib is administered for only five consecutive days at a time.

This study aimed to evaluate the usefulness of the weekends-off strategy for lenvatinib administration with respect to tolerability toward AEs and recovering of the therapeutic response. Furthermore, we evaluated the change in vascularity in both the tumor and organs using a mouse hepatoma xenograft model.

## 2. Results

### 2.1. Patient Characteristics

The clinicodemographic characteristics of the patients are summarized in Table 1. In total, 74 and 60 patients weighed ≥60 kg and <60 kg, respectively. There were 98 patients with Child-Pugh (C-P) score 5, 30 patients, C-P score 6, and 7 patients, C-P score 7. With respect to Barcelona Clinic Liver Cancer stage [21], 2, 81, and 52 patients had stage A, B, and C disease, respectively. Meanwhile, the TNM stage according to the Liver Cancer Study Group of Japan was II, III, IVA, and IVB in 5, 77, 5, and 48 patients, respectively [5]. The median tumor size was 31 mm (range, 10–170 mm), and the median alpha-fetoprotein level was 32.1 ng/mL (range, 1.5–118,660 ng/mL). The median des-gamma carboxyprothrombin level was 174.0 mAU/mL (range, 11.5–524,068 mAU/mL). The initial lenvatinib dose was 4, 8, and 12 mg in 6, 84, and 45 patients, respectively.

### 2.2. Therapeutic Effects and AEs

The median PFS of the 135 patients was 5.4 months (Appendix A). The therapeutic responses to lenvatinib are shown in Appendix A. The ORR was 37%, and the disease control rate (DCR) was 80%. Treatment-related AEs are shown in Figure 1. The incidence rates of any grade and grade ≥ 3 AEs were 82.1% and 49.6%, respectively. The dose reduction rate was 85.9%, while the treatment discontinuation rate was 57.8% (Figure 1A). The most frequent AE was hypertension (HT) followed by general fatigue. Most cases of HT were effectively managed using antihypertensive agents. General fatigue was the most common reason for dose reduction, temporary interruption, and discontinuation of lenvatinib treatment (Figure 1B).

### 2.3. Mechanisms for the Occurrence of Fatigue in Lenvatinib Treatment

Lenvatinib-induced fatigue was the most clinically critical AE. Figure 2A shows the incidence rate of lenvatinib-induced fatigue. The incidence rate of any grade and grade ≥ 3 fatigue was 53.1% and 16.0%, respectively (Figure 2A). Based on previous reports that hypothyroidism and hypoadrenocorticism are correlated with fatigue related to anti-angiogenic drugs (AAD) [17,22,23,24], we assessed the patients’ levels of free T4, thyroid-stimulating hormone (TSH), cortisol, and acetylthiocholine iodide (ACTH). Among the 135 patients, thyroid function was evaluated before and after treatment in 118 patients. The characteristics of these patients are shown in Appendix A. Overall, 12% of patients showed a decreased free T4 level, which suggested hypothyroidism, after lenvatinib treatment. Meanwhile, 60% of patients showed an elevated TSH level, which suggested subclinical hypothyroidism. With respect to adrenal gland function, it was evaluated before and after treatment in 31 patients. The characteristics of these patients are shown in Appendix A. No patients showed a serum cortisol level below the standard value before lenvatinib treatment. There were 39% of patients who showed a decreased cortisol level after treatment. We found significant correlation between fatigue and subclinical hypothyroidism (Figure 2B). In contrast, there was no significant correlation between fatigue and an increased ACTH level (Figure 2C).

### 2.4. Evaluation of Vascular Structural Change in Mouse Hepatoma Xenograft Model

We then analyzed changes in tumor vasculatures in the mouse hepatoma xenograft model treated with lenvatinib. Notably, we observed the reduction in tumor vascular density by more than 60% in response to lenvatinib (Appendix A). We also analyzed the change in vascular structure in the thyroid and adrenal gland, which might correlate with the development of fatigue in lenvatinib treatment. Interestingly, we observed a reduction in vascular density of almost 50% and 70% in the thyroid and adrenal glands, respectively, in the lenvatinib group (Appendix A). These results were consistent with our previous report in which anti-VEGF antibody was used as AAD [14]. These data demonstrate that the anti-angiogenic effect of lenvatinib is not limited to only the tumor vessels, but also the vascular structure of healthy organs.

### 2.5. Simulation of Blood Concentration of Lenvatinib in Weekends-off Protocol

The results of the simulation of the drug blood concentration for the weekends-off administration of lenvatinib are shown in Figure 3. Assuming a Cmax of 12 mg of lenvatinib, the predicted plasma concentration of lenvatinib was 18% and 1.5%, 24 hours and 48 hours after administration of 12 mg lenvatinib, respectively. According to this simulation, the biological inhibitory effects of lenvatinib would disappear after two days of no treatment. Additionally, the area under the blood concentration-time curve (AUC) of 8 mg daily lenvatinib was only 45% of 12 mg daily administration. In contrast to the simulation, the AUC of 12 mg weekends-off administration was retained at 70% of the AUC for consecutive daily administration of 12 mg lenvatinib.

### 2.6. Verification of the Usefulness of “Weekends-off” Administration of Lenvatinib

Among the 135 patients treated with lenvatinib, 30 patients were treated using the weekends-off strategy. An overall flowchart of the 135 patients treated with lenvatinib in this study is shown in Appendix A. Of the 135 patients, 20 patients (14.8%) received the initial dose of lenvatinib without any dose reduction, while the other 115 patients (85.2%) needed a dose reduction. Of the 115 patients, 28 patients received the first reduced dose and 57 patients received the second reduced dose. The weekends-off protocol of lenvatinib was administered to 30 patients. The patient and tumor characteristics are shown in Appendix A. Fatigue was the most common cause of dose reduction (Appendix A). With respect to tolerability toward AEs, 66.7% of patients treated using the weekends-off strategy were able to continue without developing intolerable AEs (Figure 4A). With respect to a therapeutic response, 80.8% of patients showed progressive disease after dose reduction of lenvatinib, which meant only the reduced dose administration of lenvatinib effected 19.2% of patients in total. However, 61.5% of patients who switched to the weekends-off protocol using the original dose recovered the therapeutic response (Figure 4B). These results suggested that a fixed proportion of patients whose lenvatinib dose was reduced due to the appearance of AEs is salvaged by weekends-off administration with respect to a therapeutic response and tolerability toward AEs. According to these results, the median administration period of lenvatinib in the patients who administered a weekends-off protocol (438 days) was significantly longer than that in the patients who could not administer a weekends-off protocol (213 days) (*p* < 0.001, Figure 4C). Additionally, we analyzed the overall survival (OS) in both groups (Figure 4D). The median survival time (MST) of patients who received the standard (continuous daily administration) protocol was 15.2 months, while >50% of the patients who received the weekends-off protocol of lenvatinib survived for at least 20 months, and their follow-up is continuing. Administration of the weekends-off protocol significantly contributed to prolonging survival in the patients with HCC who received lenvatinib treatment (*p* < 0.05). Additionally, we also compared the OS between patients who were placed on the “weekends-off” protocol and patients who did not need any dose reduction of lenvatinib. There was no significant difference between the two groups, and this confirms the effectiveness of the “weekends-off” protocol of lenvatinib (Appendix A). A representative case treated by the weekends-off protocol is shown in Figure 4E–H. This patient had multiple recurrent nodules in the liver after hepatic resection, hepatic arterial infusion chemotherapy, transcatheter arterial chemoembolization, sorafenib, and regorafenib. The initial lenvatinib dose was 8 mg once daily (Figure 4E). One month after the start of treatment, the patient was determined to have achieved partial response (PR) based on the findings of impaired enhancement of most lesions on the computed tomography (CT) (Figure 4F). However, the patient developed Grade 3 fatigue. Therefore, the dose was reduced to 4 mg daily after a temporary interruption. The fatigue was manageable, but enhanced CT showed multiple re-enhanced nodules (Figure 4G). Then, treatment was switched to the weekends-off at the original dose of 8 mg of lenvatinib. The re-enhancement disappeared again with manageable AEs (Figure 4H). In this patient, long-term lenvatinib at more than 18 months was administered under control of tumor progression.

### 2.7. Evaluation of Change of Vascular Structure in the Weekends-Off Administration of Lenvatinib

Clinically, the weekends-off method of lenvatinib was useful. To evaluate changes in vascular structure under weekends-off administration of lenvatinib, we assessed the vascular structure of both the tumor and the organs in vehicle treatment, lenvatinib 5 days-on (The mice were sacrificed on day 5), lenvatinib 7 days-on (The mice were sacrificed on day 7), and lenvatinib 5 days-on/2 days-off (weekends-off/The mice were sacrificed on day 7) groups. The weekends-off group showed significant recovery of the vascular structure in the thyroid and adrenal glands comparing with a 5 days-on and a 7 days-on group (Figure 5A,B). However, the weekends-off group also showed revascularization in the tumor (Figure 5C). These results showed that the vasculatures of not only the tumor but also of the organs dynamically fluctuated from the administration and temporary withdrawal of lenvatinib.

## 3. Discussion

In this study, we confirmed that fatigue is the most important AE that leads to the discontinuation of lenvatinib treatment among patients with HCC. The mouse hepatoma orthotopic model revealed that lenvatinib treatment reduces not only tumor vasculature, but also the vasculature of the thyroid and the adrenal glands. We proposed a weekends-off schedule of lenvatinib administration to manage lenvatinib-induced AEs while keeping the therapeutic effects. Of the patients who were treated using the weekends-off strategy, 66.7% were able to continue without developing intolerable AEs. Moreover, 61.5% of patients who switched to the weekends-off protocol using the original dose experienced a therapeutic response. The weekends-off protocol of lenvatinib also significantly prolonged the administration period and the survival time of patients with HCC treated with lenvatinib. Lastly, the mouse hepatoma orthotopic model revealed that the two days off in the weekends-off protocol allowed the vasculature of the thyroid and adrenal glands to recover, which might have contributed toward the improved tolerability toward lenvatinib-induced AEs.

MTAs including lenvatinib generally show a high incidence rate of AEs. In this study, we investigated the incidence rate, grade, and profile of AEs in lenvatinib. Of the 135 patients, only 14.8% patients tolerated the initial dose of lenvatinib without any dose reduction. In other words, in 85.2% of patients, the dose of lenvatinib had to be reduced. Furthermore, 57.8% of patients could not continue lenvatinib treatment. These results show that continuing lenvatinib treatment over a long-term period is difficult. The most common AE was hypertension, but it was manageable using antihypertensive agents. The second most common AE was fatigue, and this was the most common cause of dosage reduction, temporary interruption, or treatment discontinuation. Fatigue is a very subjective AE, and the management of subjective AEs is challenging [22]. Our results are in line with our previous study on lenvatinib. Using a mouse model, we have previously reported that inhibition of the vascular endothelial growth factor (VEGF) signal using an anti-VEGF antibody was correlated with the appearance of AEs in AADs [14]. Endocrine organs such as the thyroid and adrenal glands were the most sensitive organs against inhibition of the VEGF signal [14]. Similarly, fatigue was one of the most important AEs in previous real-word studies [25,26,27]. However, fatigue did not rank the second most common in the randomized phase III trial of lenvatinib (i.e., REFLECT study) [10,28]. This discrepancy between real-world and clinical trial data may be due to the patients’ characteristics. Specifically, the REFLECT trial enrolled patients with better liver function and similar characteristics of patients and tumors. Supportive drugs to control AEs such as hypertension will expand the use of lenvatinib. With respect to fatigue, several reports recommend the use of thyroid hormone preparations or steroids [24,29]. In addition, in our study, 20 patients required administration of thyroid hormone preparations to combat fatigue caused by lenvatinib treatment. In 16 of these patients, serum thyroid-stimulating hormone (TSH) levels decreased as a result, which indicates improvement of subclinical hypothyroidism. However, translation of this finding into overt improvement of fatigue was noted in only two patients. These results suggest that the effect of these supportive drugs varies individually and that not all patients experience improvement. Therefore, modification of the treatment regimen, such as the “weekends-off” protocol, is important for the management of fatigue during lenvatinib treatment. Additionally, although there was no significant difference between elevation of the ACTH level and fatigue in our study, one report has described adrenal insufficiency associated with development of fatigue during lenvatinib treatment [30]. In our “in vivo” study, there was a change in the vascular structure of the adrenal glands. The lack of any correlation in the present study might be explained by the fact that the number of patients in whom adrenal function was evaluated was small. Moreover, any change in metabolism of the drug because of impaired hepatic function may increase the probability of adverse effects. According to Hussein and Ikeda’s reports, the Cmax of lenvatinib is increased in patients with hepatic impairment [19,31]. However, renal impairment has no effect on the pharmacokinetics of lenvatinib. Hence, modification of the administration schedule is important to continue lenvatinib treatment.

In many cases, tumor revascularization is induced after dose reduction of lenvatinib. Additionally, in our study, 30 patients treated with the weekends-off strategy showed tumor revascularization after the dose reduction of lenvatinib. However, among the patients who required dose reduction due to AEs, a therapeutic response has also recovered again in 61.5% of the patients when the weekends-off administration was implemented using the original dose of lenvatinib. A recent study showed that Cmax and relative dose intensity (RDI) are important in the therapeutic effect of lenvatinib [32,33,34]. Ikeda et al. and Tamai et al. reported that the Cmax of lenvatinib is reduced by nearly 50% and the AUC is decreased by 55% when the dose is reduced from 12 mg to 8 mg [19,20]. This indicates that maintaining a higher dose is important to maintain a higher Cmax. Maintaining more than 70% RDI in lenvatinib has been reported to be an independent factor for a better therapeutic response and better PFS [32]. In the weekends-off protocol, the RDI can be maintained at 71.4%, and this might have, in part, accounted for the 50% improvement of a therapeutic effect in our study. According to Ikeda’s report and our data, a large dose reduction should be avoided to maintain the Cmax of lenvatinib. Therefore, the weekends-off protocol is a useful compromise. In our study, some cases that were intolerant to the weekends-off protocol of lenvatinib received every two days protocol daily or a weekends-off protocol at the second reduced dose. Modification of the administration schedule, such as every two days, at the higher dose of lenvatinib may help preserve an adequate Cmax of lenvatinib. It is crucial to find the most suitable dose and administration schedule of lenvatinib for each patient for long-term administration of lenvatinib.

Our results using a mouse hepatoma orthotopic model indicated that two days-off in lenvatinib administration resulted in revascularization of the thyroid and adrenal glands. The revascularization on withdrawal of AADs is induced by increased VEGF level in the organs, which was proven using a VEGF receptor knockdown mouse model in a previous study [15]. However, we also found that not only the organs, but also the tumor was revascularized during the weekends-off periods, which indicates that long-term rest as the potential to enable tumor regrowth. Figure 5 showed that tumor vessels were more sensitive toward AAD than other healthy organs in terms of the reduction of the vascular density. Additionally, although the vascular structure of the other healthy organs immediately recovered during the two days-off period in patients in the weekends-off group, the reduction in the tumor vascular density was sustained. Therefore, balancing the therapeutic effect and the development of AEs is important for ensuring a sustained therapeutic response to long-term lenvatinib treatment. We previously reported that discontinuation of AAD promoted cancer metastasis [15]. In our previous study, discontinuation of AAD induced structural change of the liver sinusoidal vasculature, which led to hyperpermeability and enlargement of the pore size of the sinusoidal endothelium. The VEGF blockade using anti-VEGF antibody led to a marked reduction in the hepatic vascular density, and discontinuation of the VEGF blockade led to rapid revascularization and recovery to the non-treated level. This indicates that the vasculature of healthy organs was also dynamically affected by AAD treatment. These results of the current study are consistent with the results of our previous study. Other studies have also provided evidence that AAD affects not only tumor vessels, but also those of other healthy organs [13,35].

This study has some limitations including the retrospective design and the small sample size. In addition, we only tried the weekends-off protocol and did not try other protocols, such as 3 days-on/1 day-off. There might be other better protocols for modifying the administration schedule of lenvatinib. The most important point in this study is that modifying the administration schedule is important for long-term administration of lenvatinib treatment. Therefore, the most suitable dose and administration schedule for each patient should be determined. Additionally, although we compared the administration period and OS between the weekends-off group and the other group in this study, there is a possibility that tumor and patient characteristics were different between the two groups. Further accumulation of clinical cases and experience of the usefulness of the weekends-off protocol of lenvatinib is needed. To prove the usefulness of the weekends-off protocol of lenvatinib, a prospective randomized clinical trial of the weekends-off protocol of lenvatinib should be conducted. Furthermore, although our basic experiment showed that lenvatinib led to regression of the vascular structure of the thyroid and adrenal glands, the mechanism for lenvatinib-induced fatigue is still unclear. Furthermore, the known anti-angiogenic effect of lenvatinib may not be the only mechanism that mediates its anti-neoplastic efficacy and its potential to cause AEs. Further studies are needed to understand the pharmacodynamics and pharmacokinetics of lenvatinib.

## 4. Materials and Methods

### 4.1. Study Design and Patients

This retrospective study was approved by the Ethical Committee of the Kurume University School of Medicine and was conducted in accordance with the Declaration of Helsinki (Ethical code: 18146). Written informed consent for lenvatinib treatment was obtained from each patient. We evaluated 152 patients with unresectable HCC, who were treated with lenvatinib at Kurume University School of Medicine (Fukuoka, Japan) and its affiliated hospital (Kurume Liver Cancer Study Group) between March 2018 and September 2019. Of them, 17 patients were excluded because of incorrect data. Thus, 135 patients were included in the analysis (Appendix A). Among them, 30 patients were treated with the weekends-off strategy.

### 4.2. Simulation of the Blood Concentration of Lenvatinib in the Weekends-Off Protocol

We simulated the blood concentration in the weekends-off protocol using the values reported by Ikeda et al. [19]. They reported that 12 mg and 8 mg of lenvatinib resulted in a Cmax of 349 and 167 ng/mL, respectively, and an AUC of 3960 and 1770 ng/mL/h, respectively. The blood concentration 24 hours after administration (C_24h_) of 12 mg and 8 mg of lenvatinib was 64.0 and 24.9 ng/mL, respectively. The time to the peak steady state drug concentration (Tmax) after administering 12 mg and 8 mg of lenvatinib was 2 and 2.03 hours, respectively. The terminal elimination phase half-life of 12 mg and 8 mg of lenvatinib was 9.74 and 10.2 hours, respectively. We simulated the drug blood concentration for the weekends-off administration of lenvatinib using these numerical values.

### 4.3. Weekends-off Protocol

The initial lenvatinib dose was 12 and 8 mg once daily for patients weighing ≥60 kg and <60 kg, respectively [20]. Weekends-off administration involved a cycle of five consecutive days-on and two consecutive days-off. For patients who developed unacceptable lenvatinib-related AEs (i.e., grade ≥ 3), the dose of lenvatinib was reduced or the administration was temporarily interrupted until the AEs were improved to grade 1 or 2. For patients whose AEs were managed via dose reduction, the dose was increased when the AEs were resolved. Dose reduction was based on the manufacturer’s recommendation. Specifically, for patients who received 12 mg of lenvatinib, the dose was first reduced to 8 mg and then to 4 mg. For patients who received 8 mg of lenvatinib, the dose was first reduced to 4 mg and then to 4 mg divided into two administrations per day (Appendix A). The treatment protocol is described in detail in Appendix A.

### 4.4. Assessments

Therapeutic response was evaluated using imaging tests according to the modified Response Evaluation Criteria in Solid Tumors (mRECIST) guidelines [36]. The objective response rate (ORR) was assessed as a complete response (CR) + partial response (PR). The disease control rate (DCR) was assessed as ORR + stable disease (SD). Meanwhile, AEs were assessed according to the National Cancer Institute’s Common Terminology Criteria for Adverse Events (CTCAE) version 4.0. The ratio of any grade and grade ≥3 AEs was calculated.

### 4.5. In Vivo Study

#### 4.5.1. Cell Line and Animals

Mouse hepatoma cell lines Hep-55.1C were originally purchased from CLS Cell Lines Service GmbH (Oppenheim, Germany). Cells were maintained in Dulbecco modified Eagle medium (Gibco Invitrogen Cell Culture Co, Auckland, New Zealand) supplemented with 10% fetal bovine serum (Biowest, Nuaille, France). Female 6-week-old c57BL/6 mice were purchased from Kyudo KK (Fukuoka, Japan). The mice were caged in a group of six or fewer mice per cage at the animal facility of Kurume University School of Medicine. Mice were sacrificed via cervical dislocation after anesthesia using isoflurane. All applicable international, national, and/or institutional guidelines for the care and use of animals were followed, and all animal experiments were approved by the ethical committee of the Kurume University School of Medicine (Ethical code: 2018-191-1).

#### 4.5.2. Lenvatinib Treatment of Mouse Hepatoma Orthotopic Model

Lenvatinib (Funakoshi Co., Ltd., Tokyo, Japan) was dissolved in 3 mmol/L HCL. A total of 2 × 10^6^ Hep55.1-c cells were inoculated into the left lobe of the mice livers. After 10 days from inoculation of tumor cells, the mice were randomly allocated to the treatment groups and was orally given 10 mg/kg of lenvatinib or the corresponding vehicle. As a standard administration schedule, lenvatinib was administered once daily. For a weekends-off administration schedule, lenvatinib was administered for five consecutive days-on and two consecutive days-off. Each group was comprised of 6 mice. After the mice were sacrificed, the tumor, adrenal glands, and thyroid glands were collected and weighed. They were then immediately fixed with 4% paraformaldehyde (PFA) overnight, which was followed by washing with phosphate buffered saline (PBS).

#### 4.5.3. Whole-Mount Staining

Whole-mount staining was performed according to our previous reports [37,38]. Tissues were cut into thin slices and fixed in 4% PFA overnight, and then exposed to 20 μg/mL proteinase K. Thereafter, the tissues were incubated with goat anti-CD31 antibody (R&D System Inc, Cat. No. AF-3628) overnight at 4 °C, which was followed by staining with a donkey anti-goat secondary antibody (Abcam, Tokyo, Japan) for 2 hours at room temperature. Slides were mounted and examined under a microscope (Keyence, Osaka, Japan). We scanned 5 thin sections at 3-μm distances of each sample and projected three-dimensional images of each tissue sample. Quantitative analyses from at least six different sections were performed using the Adobe Photoshop CS software program (Adobe systems, Tokyo, Japan).

#### 4.5.4. Statistical Analysis

All statistical analyses were performed using JMP statistical analysis software (JMP Pro version 14, Tokyo, Japan). OS and PFS were calculated using the Kaplan-Meier method and analyzed using the log-rank test. All experimental data were expressed as mean ± standard deviation (SD). Between-group comparisons were performed using the Mann-Whitney U test, the Kruskal-Wallis test, and nonparametric analysis of variance. If the one-way analysis of variance was significant, differences between individual groups were analyzed using the Fisher least significant difference test. *p* < 0.05 was considered statistically significant.

## 5. Conclusions

A weekends-off protocol for lenvatinib administration improves in controlling AEs while maintaining the optimal Cmax and RDI. Thus, a weekends-off protocol not only improves tolerability toward AEs, but also improves the therapeutic response. Our results provide clinically important information in lenvatinib treatment.

## Figures and Tables

**Figure 1 cancers-12-01010-f001:**
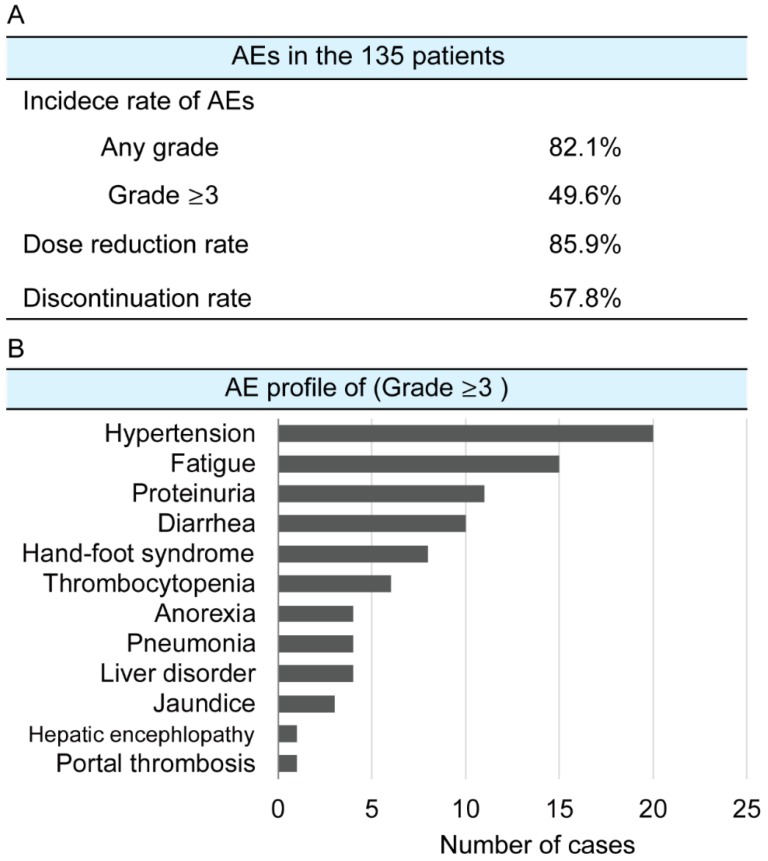
Adverse events of lenvatinib treatment. (**A**) Incidence rate, dose reduction rate, and treatment discontinuation rate. (**B**) Profile of adverse events. AE: adverse event.

**Figure 2 cancers-12-01010-f002:**
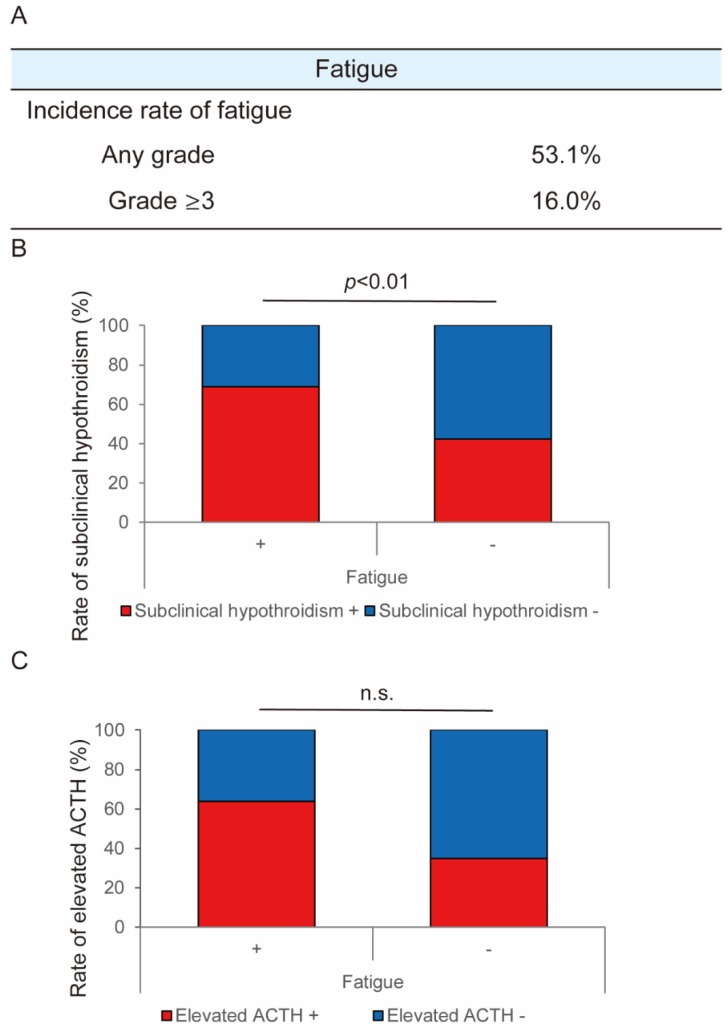
Analysis of lenvatinib-induced fatigue. (**A**) Incidence rate. (**B**) Correlation between fatigue and thyroid function (fatigue was significantly correlated with subclinical hypothyroidism). (**C**) Association between fatigue and adrenal function (fatigue was not significantly correlated with elevated acetylthiocholine iodide (ACTH)). n.s.: not significant.

**Figure 3 cancers-12-01010-f003:**
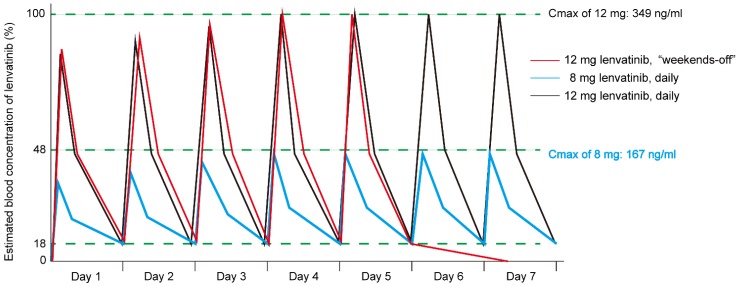
Simulation of the blood concentration of lenvatinib. The blood concentration curve for each administration method (red: 12 mg “weekends-off” method, blue: 12 mg daily, black: 8 mg daily) was simulated according to the pharmacokinetics study of lenvatinib. The Cmax of 12 mg and 8 mg daily lenvatinib is 349 ng/mL and 167 ng/mL, respectively. The blood concentration of lenvatinib after 48 hours upon the last dose is predicted to be 1.5% of Cmax. The area under the blood concentration-time curve (AUC) of the weekends-off method is predicted to be 70% of that achieved with daily lenvatinib administration.

**Figure 4 cancers-12-01010-f004:**
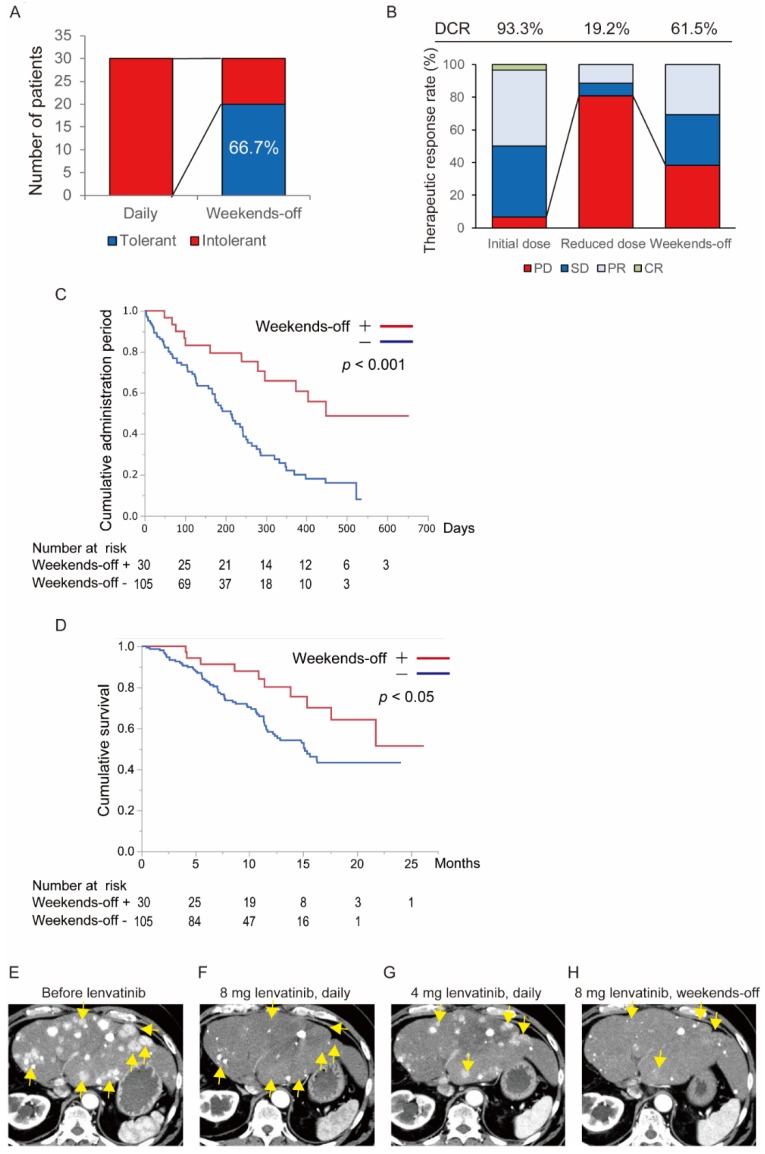
Assessment of the weekends-off administration of lenvatinib. (**A**) Tolerability toward adverse events (AEs) (66.7% of patients were tolerant to lenvatinib-induced AEs). (**B**) Therapeutic response according to the type of lenvatinib protocol. The disease control rate (DCR) of patients treated with lenvatinib at the initial daily dose was 93.3%. After dose reduction due to the appearance of adverse events, the DCR was reduced to 19.2%. After switching to the weekends-off protocol, the DCR increased to 61.5%. (**C**) Duration of lenvatinib administration in 30 patients treated using the weekends-off protocol (red line) and 105 patients treated using the continuous daily administration protocol (blue line). The median administration period of patients who were treated with the weekends-off protocol was significantly longer than that of patients treated with the continuous protocol (*p* < 0.001). (**D**) The survival curve of patients treated with lenvatinib. The red line shows the survival curve of patients who followed the weekends-off protocol and the blue line shows the survival curve of patients who followed the standard (continuous daily administration) protocol. The median survival time (MST) of patients who followed the standard protocol was 15.2 months, while >50% of the patients who received the weekends-off protocol of lenvatinib survived for at least 20 months, and their follow-up is continuing. The MST of patients who followed the weekends-off protocol of lenvatinib was significantly longer than that of patients who followed the standard protocol (*p* < 0.05). (**E**–**H**) Representative computed tomography (CT) images of patients who received weekends-off administration of lenvatinib. (**E**) CT image before administration of lenvatinib. Multiple enhanced lesions are shown (arrows). (**F**) CT image after 8 mg of lenvatinib once daily. Enhancement of most lesions disappeared (arrows). (**G**) CT image after dose reduction to 4 mg of lenvatinib once daily. Most lesions showed re-enhanced vascularity (arrows). (**H**) CT image after weekends-off administration of lenvatinib. Tumor vascularity was reduced again (arrows). Abbreviations: AEs, adverse events, CR, complete response, CT, computed tomography, DCR, disease control rate, PD, progressive disease, PR, partial response, SD, stable disease, MST, median survival time.

**Figure 5 cancers-12-01010-f005:**
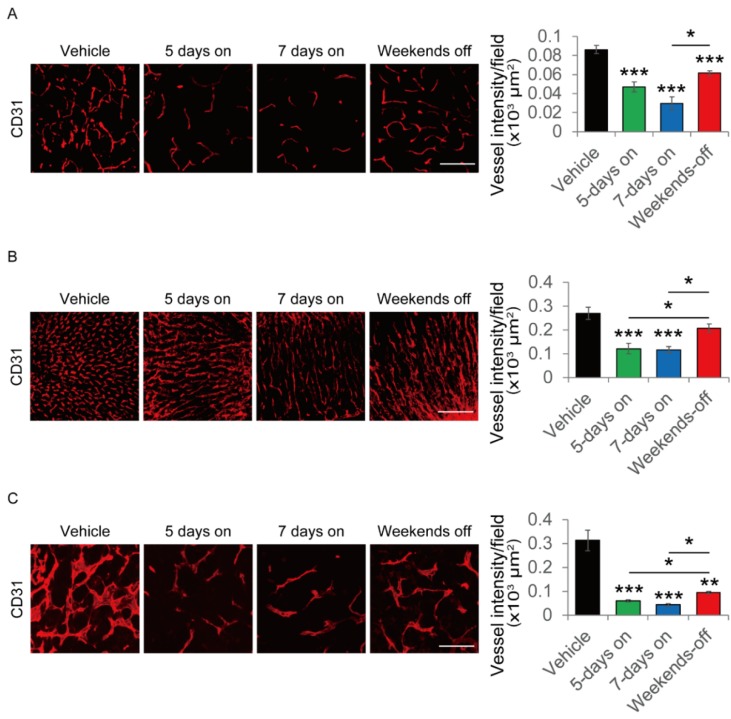
Assessment of vascular change in the weekends-off method of lenvatinib administration in the mouse hepatoma orthotopic model. (**A**) Vascular change in the thyroid. Quantification of CD31+ tumor vessels (six random field from three independent tumor samples per group). (**B**) Vascular change in the adrenal glands. Quantification of CD31+ tumor vessels (six random field from three independent tumor samples per group). (**C**) Vascular change in the tumor. Quantification of CD31+ tumor vessels (six random field from three independent tumor samples per group). Bar represents 200 μm. * *p* < 0.05, ** *p* < 0.01, *** *p* < 0.001. Data are presented as means ± SEM. SEM: standard error of the mean.

**Table 1 cancers-12-01010-t001:** Baseline clinical demographic and tumor characteristics of the patients (*n* = 135 patients).

Clinicodemographic Characteristics	Value
Age (years)	74 (44–89)
Sex Male/Female	113/22
Etiology HBV */HCV **/non-B, non-C	27/63/45
Child-Pugh score 5/6/7	98/30/7
ALBI grade *** 1/2	55/79
Body weight Less than 60 kg/over 60 kg	61/74
Tumor characteristics	
BCLC ^†^ stage A/B/C	2/81/52
TNM ^††^ stage II/III/IVA/IVB	5/77/5/48
Tumor size (mm), median (range)	31 (10–170)
Alpha-fetoprotein (ng/mL)	32.1 (1.5–118,660)
DCP ^†††^ (mAU/mL)	174.0 (11.5–524,068)
Lenvatinib	
Initial dose (mg) 4/8/12	6/84/45

* Hepatitis B virus; ** Hepatitis C virus; *** Albumin-bilirubin grade, ^†^ Barcelona clinic liver cancer, ^††^ Tumor, node and metastasis, ^†††^ Des-gamma carboxyprothrombin.

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
