# Peer review of "Weekends-Off Lenvatinib for Unresectable Hepatocellular Carcinoma Improves Therapeutic Response and Tolerability Toward Adverse Events"

_cancers, 2020, doi:10.3390/cancers12041010_

Round 1
Reviewer 1 Report
In the manuscript, the authors proposed “weekends-off” lenvatinib as a useful protocol to recover the therapeutic efficacy and tolerability of adverse events. This article is of interest and with important messages; however, there are several problems in this manuscript, and a significant revision is needed before ready for publication.
Major points:
1) Lenvatinib-induced general fatigue is the most clinically critical AE and is the main reason for dose reduction and treatment interruption. The authors described that of the 118 patients evaluated thyroid function, 12% of patients showed hypothyroidism, and 60% of patients showed subclinical hypothyroidism. The authors also found a significant correlation between fatigue and subclinical hypothyroidism. How many people received thyroid hormone preparations? Are there any patients who did not need dose reduction, “weekends-off” protocol, or treatment interruption by the use of thyroid hormone preparations or steroids? Are there any patients who could return to regular dose or restart treatment by the use of thyroid hormone preparations?
2) The authors reported that among 135 patients treated with lenvatinib, 30 patients were treated using the weekends-off strategy. How was the treatment course of the other 105 patients? How many patients experienced dose reduction, treatment interruption, or not? Are there any patients who returned to regular dose after weekends-off protocol? Please show the flowchart.
3) When a patient could not tolerate 8mg lenvatinib weekends-off protocol, was the treatment switched to 4mg daily or 4mg weekends-off protocol?
4) The authors showed the simulation of blood concentration of lenvatinib, which is an essential part of this article. How much is this simulation affected by the patient physique and liver/renal function?
5) Is the “weekends-off” protocol clinically best and superior to other protocols, such as ‘3days-on and 1day-off’?
Minor points:
1) In the top line of Supplementary Table 1, Clinicodemographic characteristics of “30” patients?
2) The numbers of subjects at risk should be shown with the survival curves in Figure 4C/D and Supplementary Figure 4A.
Reviewer 2 Report
This is a manuscript entitled “Weekends-off lenvatinib for unresectable hepatocellular carcinoma improves therapeutic response and tolerability toward adverse events
” by Iwamoto, et al.
In this study, the authors aimed to clarify the AEs of lenvatinib and the therapeutic impact of 5 days-on/2 days-off administration in clinical study and in vivo analysis
The authors showed that of the 30 patients who received weekends-off lenvatinib, 66.7% tolerated the AEs. Although 80.8% of the patients showed progression after dose reduction, the therapeutic response improved in 61.5% of the patients by weekends-off lenvatinib. Notably, weekends-off administration significantly prolonged administration period and survival. The mouse hepatoma model showed that weekends-off administration contributed to recovery of vascularity in the organs.
The management of AEs in lenvatinib is clinically important issue. However, in this study, how patients who received the Weekends-off were selected, is vague, the included number of those patients was small, and those patients’ baseline characteristics seemed to be different from the whole cohort. Thus, because there seems to be a big bias, it seems to be difficult to conclude that Weekends-off administration of lenvatinib was useful to recover the therapeutic response and tolerability toward AEs.
In addition, OS of those patients seemed to be quite excellent, thus, please compared with patients who did not need dose reduction.
In addition, this reviewer concerns below.
- In Figure 2. The number of patients who were evaluated with ACTH was quite small compared with that with hypothyroidism, thus, the reviewer think that it is hard to conclude that ACTH might be associated with fatigue.
- Please show the effect of Weekends-off lenvatinib on tumor size, because lenvatinib has not only anti-vascular effect, but also direct anti-tumor effect, and compared with 5 days on and 7 days on.
Round 2
Reviewer 1 Report
The manuscript has been much improved and is in a nice condition now.
I think this manuscript is acceptable for publication.